# Predictors of Oral Health Behaviors among Dental Students

**DOI:** 10.3390/medicina59010106

**Published:** 2023-01-03

**Authors:** Adina Oana Armencia, Ramona Feier, Vlad Dănilă, Dana Gabriela Budală, Carina Balcoș, Dana Baciu, Marius Prelipceanu, Dragoș Ionuț Vicoveanu

**Affiliations:** 1Surgery I Department, Faculty of Dental Medicine, “Gr. T. Popa” University of Medicine and Pharmacy, 700115 Iasi, Romania; 2Faculty of Medicine, “Dimitrie Cantemir” University, 700115 Iasi, Romania; 3Prosthodontics Department, Faculty of Dental Medicine, “Gr. T. Popa” University of Medicine and Pharmacy, 700115 Iasi, Romania; 4Department of Computers, Electronics and Automation, Faculty of Electrical Engineering and Computer Science, “Stefan cel Mare” University of Suceava, 720225 Suceava, Romania

**Keywords:** Health-Promoting Lifestyle Profile II -HPLP II questionnaires, Generalized Self-Efficacy Scale questionnaires, oral health, planned behavior model

## Abstract

*Background and Objectives:* College life is a time when students take more responsibility for health-related behaviors. *Aim:* To determine the oral health behaviors of dental students, to motivate their transformation into cyanogenic behaviors by applying the planned behavior model, and to determine the degree to which students can modify behavior for oral health. *Material and Methods:* The results of the initial and final assessment (4 months apart) of the bacterial plaque present by means of the Quigley Hein and API indexes were collected from the student files. The Health-Promoting Lifestyle Profile II -HPLP II and Generalized Self-Efficacy Scale (GSE) questionnaires were used to determine the variables with predictor values that influence the cognitive-perceptual factors. *Results:* For the “health responsibility” domain, the average total value was 2.53 ± 0.9 (SD). For the “interpersonal relationships” domain, the average value was 2.82 ± 0.79. In the case of the “nutrition” domain, the desire to change the behavior of the participants was moderate (2.53 ± 1.19). “Physical activity” represents an important field in the everyday life of young adults. The results of the study indicated a moderate desire for change (2.52 ± 0.97). In the case of “spiritual growth”, in which self-esteem, the way the individual perceives the environment and the adaptation to it were evaluated, a great desire for change was observed (2.74 ± 0.82). Scale GSE questionnaire showed an α coefficient of 28.52 ± 0.82, a value that indicated a moderate to a high level of efficacy and self-confidence. *Conclusions:* The results of the study showed that students have a moderate tendency to change in all areas, except for interpersonal relationships, where the desire to change something is increased. The participants have a moderate to a high level of self-efficacy and self-confidence.

## 1. Introduction

The university environment provides the ideal setting for health promotion and education. Because students are more prone to unhealthy lifestyles (smoking, unhealthy eating, increased stress, and a sedentary lifestyle), universities are responsible for creating environments that support health and help students control their health [1,2].

Higher education has seen a rise in interest in implementing a health promotion strategy in recent years. Healthy habits and overall happiness among students can have a good effect on their academic performance and general health [3].

College life is a time when students take more responsibility for health-related behaviors. Some studies demonstrate the existing relationship between the perception of health and the responsibility for the health of young people in the university environment [4]. The way barriers and benefits to preventive health behaviors are perceived affects health responsibility. Thus, socioeconomic, and psycho-social status and integration problems in the university environment affect behaviors aimed at protecting and maintaining health [3,4,5].

Perceived stress is also integrated as a modulating component, and research suggests that it can be decreased by engaging in health-promoting behaviors and developing personal coping mechanisms. Due to the increased intellectual demands, new social ties, financial difficulties and limited support from family and friends, the transition to university can be quite stressful. Increased stress during this transition period may reduce the enactment of health-promoting behavior by reducing personal intentions to engage in health-promoting behaviors [6,7].

Social support has a direct impact on health, promoting behaviors that can improve health or lower stress and fostering behaviors by creating favorable environments and minimizing adverse situations [1,4].

Oral health behavior research has explored the effectiveness and applicability of various models in changing oral health behaviors. Health as a belief model, the planned behavior model, and the theory of motivated action are examples of models that focus on individuals taking responsibility for their own health [8,9].

The nature of the connections between ideas and attitudes is described by the theory of planned behavior. This theory holds that attitudes toward behaviors are evaluated in relation to possible personal principles about those behaviors, where a belief is defined as the possibility for behavior to be directed towards a certain outcome. In particular, the assessment of the result shows a behavior that is directly related to the person’s specific likelihood of reaching that end [10]. In the theory of planned behavior, the role of attitude control results from Bandura’s notion of self-efficacy. Fishbein and Cappella stated that self-efficacy is the same as perceived behavioral control in their integrative model [11].

One of the fundamental tactics for changing the educational system and attempting to improve the state of oral health is to establish and implement educational programs using excellent educational models and theories [12].

In the specialized literature, studies document that students usually follow unhealthy lifestyles, ignoring physical activity and responsibility for health. Some focus on the theme of self-efficacy, seen as the ability to achieve or not succeed in achieving goals. This is identified with the personal evaluation of the efficacy or competence to carry out a particular behavior successfully; it is the crucial factor in predicting a person’s behavior change, as it represents what the individual believes they can do. Although self-efficacy has a direct correlation with health behavior, it also has an indirect effect on health behavior through its influence on objectives. Self-efficacy is one of the most significant determinants of health-promoting behaviors among students, impacting their efficacy or competence to carry out a particular behavior, according to a growing body of research [7,13].

Through tenacity and commitment to serious work, ambition, adaptability, and resilience to pressure and depressive moods, self-efficacy enhances personal stamina [14,15,16,17].

For these reasons, this study aimed to determine the oral health behaviors of dental students, to motivate their transformation into healthy behavior by applying the planned behavior model, and to determine the degree to which students can modify behavior for oral health.

## 2. Materials and Methods

In order to achieve the proposed objectives, a cross-sectional descriptive study was carried out on a group of 68 students from the 5th year of the English series of the Faculty of Dental Medicine of the “Grigore T. Popa” University of Medicine and Pharmacy Iași, between October 2021 and April 2022. There were 78 students enrolled in the 5th year, English section. Applying the calculation formula for a confidence level of *p* = 95%, z = 1.96, with a margin of error of 5%, a number of 65 students should participate in the study to be relevant from a statistical point of view [18]. Since several students expressed their desire to participate in the study, we enrolled all those who wished. This time of the year was chosen to avoid stress during the exam session. This study intended to identify the usual pattern of oral health promotion behaviors. This research study was authorized by the “Grigore T. Popa” Iasi University’s Ethical Committee (No. 183/05.05.2022).

The students were informed about the purpose of the study, and it was explained that participation is voluntary and the information is confidential and anonymous. The information regarding demographic characteristics and healthy behavior as well as the results of the initial and final assessment (4 months apart) of the bacterial plaque present by means of the Quigley Hein and API indexes were completed by the student before the clinical examination. The evaluation of bacterial plaque indices was aimed at highlighting the changes in oral health behaviors.

The Quigley-Hein index quantitatively evaluates the presence of bacterial plaque on certain dental surfaces (buccal and oral) of teeth after revealing the bacterial plaque. For each surface, scores between 0 and 5 are given (0 meaning the absence of bacterial plaque on the evaluated surface, and 5 meaning more than 2/3 of the evaluated surface is covered with bacterial plaque). The score is obtained by dividing the amount by the number of evaluated surface scores obtained [19,20].

The API index (Approximal Plaque Index) qualitatively evaluates the presence of bacterial plaque in the interproximal spaces. The value 1 is given for each evaluated interproximal space. The final score is obtained by dividing the number of interproximal spaces with a bacterial plaque by the number of evaluated interproximal spaces, and then the value obtained is multiplied by 100 [21].

After the initial assessment, students received additional information on how to properly brush their teeth using the BASS technique [22] and the importance of using aids for tooth brushing (dental floss, mouthwash). The students were practically shown the correct technique for brushing their teeth and how to use interdental floss. A PowerPoint presentation was also used in which the subjects received information related to brushing and adjuvants for tooth brushing. Thus, they were encouraged to change their health-related behavior orally by applying the principles underlying the planned behavior model. This need for change was explained by referring to an action, aim, circumstances and moment. Thus, if the goal was “I need to brush my teeth at least 2 times with correct brushing technique, every day for the next few months”, “doing” is the action, “correct brushing technique” is the target, “two brushings per day” is the circumstances, and “in the coming months” is the moment. If the goal was “must floss after meals at least 1 time/day in the next months”, “use” is the action, “floss” is the target, “after meals” is the context, and “in the next months” is the time. Since the relevant and central beliefs for a particular behavior were different depending on the person, we tried to identify the relevant behavioral elements, the influence of culture, the intermediaries, and the blockages typical to each behavior using questions such as the following:

What do you like/dislike about your behavior?

What are some disadvantages of your behavior?

Would the people around you be against your behavior?

What things prevent you from performing the right behavior?

If you want to change your behavior, how sure are you that you can?

If the students considered the suggested behavior to be positive (attitude) and those around them want them to change their behavior (subjective norm), the result is a higher intention (motivation) and an increase in the probability of implementing the new behavior.

The theoretical framework for this study was based on the Pender model of health promotion [23]. For our study, cognitive-perceptual factors included perceived self-efficacy and health value. Modifiers included age, gender, place of residence, perceived social support, and perceived stress.

The Health-Promoting Lifestyle Profile II (HPLP II) and Generalized Self-Efficacy Scale questionnaires were used to determine the variables with predictor values that influence the cognitive-perceptual factors [23,24].

The HPLP II consists of 52 items in six subscales: spiritual growth, responsibility for health, physical activity, nutrition, interpersonal relationships, and stress management. Respondents indicate the frequency with which they engage in each behavior using a four-point scale ranging from 1 (never) to 4 (currently). The score for the general health-promoting lifestyle is obtained by averaging the individual responses to all 52 items and is a maximum of 4. The HPLP II score is divided into 3 subscales: low (1.60–2.25), moderate (2.26–2.71), and high (2.72–3.27). The six subscale results are given by the responses to the subscale items. Utilizing mean scores maintains the 2-to-4 measurement of responses and permits important comparisons of scores from the subscales. This tool will enable the investigation of oral health patterns, health-promoting lifestyle determinants, and the effects of lifestyle modification interventions.

The second questionnaire, the Generalized Self-Efficacy Scale (GSE), is designed to assess both one’s own beliefs and the self-confidence needed to cope with a variety of difficult life situations. It consists of ten elements, the answer options ranging from partially true (1) to exactly true (4) [6]. For the GSE, the total score (α coefficient) varies between 10 and 40, with a higher score indicating more self-efficacy and self-confidence [12].

According to the criteria established by Bandura [25], the “constructed” results and expectations were defined as physical, social, and self-evaluative results.

All collected data were entered and analyzed with SPSS (Statistical Package for Social Sciences) 26.0. Descriptive statistical analysis used mean and standard deviation to highlight changes in health behaviors. Logistic regression was used to determine the influence of predictors for oral health. The ANOVA test was used to compare the means between assessment steps. A *p*-value < 0.05 is considered statistically significant.

## 3. Results

The study group was formed by 68 young adults, with an average age of 24.10 ± 1.351 years (minimum age of 22 years and maximum of 29 years, 69% of them being female subjects, 79.4% coming from the urban environment (Table 1).

Regarding the distribution of the answers to the questions regarding healthy attitudes, it was observed that 70.6% of the participants performed at least 2 tooth brushings per day, this being done in proportion to 57.4% through a technique recommended by the doctor; for most of the participants brushing was manual (86.8%); 52.9% of them used dental floss to clean the interproximal spaces (Table 1).

Regarding the distribution of responses to the Health-Promoting Lifestyle Profile II questionnaire, the trend of behavior change was observed in all directions. Thus, for the “health responsibility” domain, the average total value was 2.53 ± 0.9 (SD), indicating a moderate tendency to change behavior; the highest values were recorded for the variable “sometimes” and “often” for all evaluated items (Table 2).

For the “interpersonal relationships” domain, the average value of 2.82 ± 0.79 indicated a great desire for change, especially regarding “spend time with close friends” (3.22 ± 0.80) and “touch and am touched by people I care about” (3.22 ± 0.80) (Table 2).

In the case of the “nutrition” domain, the desire to change the behavior of the participants was moderate (2.53 ± 1.19). It was observed that the participants understood the message regarding the importance of nutrition in maintaining optimal oral health. The initial value was 2.53 ± 1.19, with the reassessment being 2.8 ± 1.15. The lowest desire to change was recorded in the case of cereal consumption (2.25 ± 0.85) (Table 2).

“Physical activity” represents an important field in the everyday life of young adults. The results of the study indicated a moderate desire for change (2.52 ± 0.97), with the highest desire for change being recorded for daily relaxation time (2.88 ± 0.92), and the lowest for performing stretching exercises 3 times a week (2.24 ± 0.86).

In the case of “spiritual growth”, in which self-esteem, the way the individual perceives the environment, and the adaptation to it were evaluated, a great desire for change was observed (2.74 ± 0.82), especially regarding “believe that my life has a purpose” (3.01 ± 0.83) and “look forward to the future” (3.24 ± 0.75). The lowest desire for change was recorded in the case of the desire to perform relaxation exercises (2.18 ± 0.84) and to reduce agitation to prevent fatigue (2.32 ± 0.67) (Table 2).

For the “stress management” domain, where the subjects were questioned about their ability to adapt to stressful situations, challenges and changes, a value of 2.51 ± 0.83 was recorded, a value that indicated a moderate desire to change something. The lowest values were recorded for the items “check my pulse rate when exercising” (2.18 ± 0.97) and “reach my target heart rate when exercising” (2.15 ± 0.90) (Table 2).

The analysis of the results of the Generalized Self-Efficacy Scale (GSE) questionnaire showed an α coefficient of 28.52 ± 0.82, a value that indicated a moderate to the high level of efficacy and self-confidence. The highest scores were recorded in the case of items that emphasize the subjects’ ability to solve difficult situations by finding a solution (item 9: 3.09 ± 0.748) if they put in enough effort (item 1: 3.10 ± 0.813; item 6: 3.03 ± 0.880) (Table 3).

To highlight changes in oral health behavior, the changes in some indicators (bacterial plaque) and healthy behaviors (tooth brushing frequency/day, type of brushing performed, use of dental floss) were tracked. The results of the statistical analysis revealed statistically significant changes (*p* = 0.000) in the case of the bacterial plaque indices Quigley Hein and API; in the case of these indicators, lower values were recorded at the final reassessment compared to the initial one (QH: initial 1.089 ± 0.54, final 0.74 ± 0.47; API: initial 60.17% ± 27.66, final 37.16% ± 16.20) (Table 4).

Table 5 shows the change in attitude regarding tooth brushing, in the sense of increasing the percentage of subjects who brush their teeth at least 2x/day, with the percentage of those who brush their teeth 3x/day remaining unchanged. Changes were also registered in the case of the number of users of dental floss in the sense of the increase in the number of students who use auxiliary means for tooth brushing. It was observed that the way teeth are brushed (manual or electric) did not change, probably due to convenience or the belief that the brushing technique used is the best.

Multiple regression analysis of the demographic variables (gender, place of residence) with a score of six subcategories was performed to determine the predictors (Table 6).

In the case of health responsibility, the urban residence and female gender are related to better oral health responsibility (β = 0.128, *p* = 0.005 and β = 0.092, *p* = 0.032, respectively). The female gender was a predictor for lower interpersonal relationships and physical activity (β = −0.112, *p* = 0.023, respectively β = −0.188, *p* = 0.028). The urban residence and female gender were predictors for better nutrition (β = 0.135, *p* = 0.037 and β = 0.182, *p* = 0.075, respectively). The urban residence is a predictor for better spiritual growth and stress management (β = 0.109, *p* = 0.030 and β = 0.211, *p* = 0.002) and female gender for lower spiritual growth and stress management (β = −0.113, *p* = 0.026 and β = −0.163, *p* = 0.001, respectively).

The results of the correlation analysis between the GSES items and the demographic factors show that these correlations are significant at the 0.05 level for items 6 and 8 in relation to the residence and with items 2, 3 and 6 in relation to gender (Table 7).

## 4. Discussion

This study showed that students perceive their oral status in a favorable and accurate manner, having, in the vast majority, healthy behaviors. Any misjudgment of oral health behavior can be a barrier to changing it. Individual and personality factors such as external situations influence their health behaviors. As every person lives a situation in a different, specific way, information is perceived through different perceptions, and behavior is affected by skills [25].

We can also say that the environment from which the subjects participating in the study come is not a predictor of healthy behavior, because it influenced spiritual growth and stress management only in few female subjects. In our study, the dimensions where the obtained results were moderate concerning a healthy lifestyle were responsibility for health, physical activity, nutrition, and stress management.

The striking result of this study was that dental students scored quite low on the health responsibility subscale. This is because they are probably not trained to feel responsible for their health but tend to engage in risky health behaviors. It is known that academic facilities and services are often oriented towards sedentary activities (for example, students spend a lot of time in front of the computer) and not towards physical activities (students rarely do outdoor activities).

In our study, perceived self-efficacy was a positive predictor of oral health-promoting behaviors among college students, suggesting that individuals with higher perceived self-efficacy are more likely to engage in health-promoting behaviors. Our results are supported by previous studies on the determinants of health-promoting lifestyle behaviors, which claimed that self-efficacy was a cognitive-perceptual factor that was strongly linked to behaviors that promoted health [26].

The behavioral intention in this study was discovered to be a predictor for the behavior vis-à-vis tooth brushing and the use of its auxiliary means. Tooth brushing is a behavior that is personally controllable. When young people feel in control of their behavior, they are more likely to practice good oral hygiene. They cannot engage in effective behaviors to promote oral health if they lack strong self-efficacy and accurate information [27,28].

At the present time, the results of existing studies have shown that students’ self-efficacy beliefs can become better using student-focused learning strategies such that they can solve their own problems and develop their critical thinking and communication skills [29,30].

A study by Sussman and Gifford demonstrated that students become more likely to report positive attitudes towards oral health behavior if they associate, with their motivation for change, three essential components: attitudes, social expectations, and perceived behavioral control [31].

One of the critical times in youth is during college, known as a dynamic, transitional period. In this period of time, students begin to take ownership of their physical, mental, and social well-being. They must be aware of the proper health behaviors and use them to enhance their quality of life and overall health [32].

Changes of healthy behaviors were observed in the sub-dimensions of spiritual growth and interpersonal relationships. The results demonstrated that the differences in language and culture do not represent a barrier in communication. This is possible because spiritual growth influences their life goals as well as their self-development skills.

The results of existing studies show that the increased score of the spiritual dimension is due on the one hand to the conscious planning of their goals in life, and on the other hand, to the awareness of the aspects that can be improved.

The results of our study show that gender influences the dimension related to interpersonal relationships. Thus, male students received lower scores on this subscale than female subjects, and this difference can be correlated with monthly income; consequently, people with higher family incomes scored higher on this subscale [1,29].

In the case of sub-dimensions addressing stress management, the changes were moderate, improving with age and experience. Moderate resistance to stress showed relatively increased effectiveness in increasing the level of awareness of healthy practices among students. Stress management training has been shown to be beneficial when the annual curriculum includes stress management education courses [5,33].

The results of the study showed a moderate desire to do physical activity. This finding indicated that students had a more sedentary lifestyle, influenced by various factors including low self-efficacy as well as a lack of engagement in regular physical activities. Inadequate physical activity associated with an unhealthy diet are common problems among students. Hence, the moderate desire for change obtained in these subdimensions. The conducted study highlighted a moderate to high intention to change an unhealthy behavior. Therefore, to facilitate the process of transforming intention into behavior, it must be supported both by increased self-efficacy and social support, since some daily behaviors (tooth brushing or flossing) do not depend on intention but on habits and social support. In most studies, physical activity had one of the lowest overall scores among the subcategories [5,34,35]. Between 40% and 50% of students seem to be sedentary and physically inactive [36].

Ramezankhani et al. reported that 21.9% of students flossed every day, whereas Alidosti et al. showed that 37.5% of the study’s student participants flossed on a regular basis [12,37].

However, the factors that determine and influence individual oral health behavior are largely unknown; for this reason, the effectiveness of prevention and health education classes varies greatly [38].

This study has some limitations, including the small sample size (pilot study) and subjective evaluation (individual cognitive processing drives the start of behavior modification action). In-depth studies on a larger group of participants are needed to identify and understand the reasons for these behaviors and how they can be transformed from potentially unhealthy behaviors to healthy ones.

## 5. Conclusions

The results of the study showed that students have a tendency to change in almost all areas evaluated by the HPLP II questionnaire. The level of self-efficacy and self-confidence was medium.

Statistically significant changes were found in the Quigley Hein and API plaque indices, as well as the trend in tooth brushing attitude change.

It is not enough to identify and understand the reasons for these behaviors; we must find out how they can be transformed from potentially unhealthy behaviors to healthy behaviors.

The findings of this study provide information about health-promoting behaviors and their determinants in dental students that can help in planning educational programs and developing health-promotion programs that support healthy choices among students. Perceived self-efficacy should be considered when developing an oral health promotion program among college students.

## Figures and Tables

**Table 1 medicina-59-00106-t001:** Demographic characteristics and oral hygiene habits of the participants.

Demographic Characteristics		No	%
Age	24.10 ± 1.35 years (SD) (minimum age of 22 years and maximum age 29 years)		
Sex	Female	47	69.1
	Male	21	30.9
Place of residence	Urban	54	79.4
	Rural	14	20.6
Oral hygiene habits			
Dental brushing frequency/day	1	3	4.4
	2	48	70.6
	3	17	25.0
Recommended tooth brushing technique	no	39	57.4
	yes	29	42.6
Type of toothbrushing	manual brushing	59	86.8
	electric brushing	9	13.2
Use of dental floss	no	32	47.1
	yes	36	52.9

**Table 2 medicina-59-00106-t002:** The distribution of the answers to the Health-Promoting Lifestyle Profile II questionnaire, the average value, and the SD of the score.

Domain	Questionnaire Item	Question	Variables				Mean Value ± SD	Total Value ± SD for Each Subscale
			Never %	Sometimes %	Often %	Routinely %		
health responsibility (nine items)	1	Report any unusual signs or symptoms to a physician or other health professional.	10.3	33.8	45.6	10.3	2.56 ± 0.81	2.53 ± 0.90 (Moderate)
	2	Read or watch TV programs about improving health.	13.2	42.6	27.9	16.2	2.74 ± 0.82	
	3	Question health professionals in order to understand their instructions.	4.40	44.1	44.1	7.40	2.54 ± 0.70	
	4	Get a second opinion when I question my healthcare provider’s advice.	8.80	35.3	38.2	17.6	2.65 ± 0.87	
	5	Discuss my health concerns with health professionals	10.3	44.1	35.3	10.3	2.46 ± 0.81	
	6	Inspect my body at least monthly for physical changes/danger signs.	7.40	39.7	39.7	13.2	2.59 ± 0.81	
	7	Ask for information from health professionals about how to take good care of myself.	8.80	38.2	41.2	11.3	2.56 ± 0.81	
	8	Get support from a network of caring people.	20.6	30.9	33.8	14.7	2.43 ± 0.98	
	9	Attend educational programs on personal health care.	19.1	42.6	30.9	7.40	2.26 ± 0.85	
interpersonal relationships (nine items)	1	Discuss my problems and concerns with people close to me.	1.50	30.9	41.2	26.5	2.93 ± 0.79	2.82 ± 0.79 (High)
	2	Praise other people easily for their achievements.	4.40	25.0	54.4	16.2	2.82 ± 0.75	
	3	Maintain meaningful and fulfilling relationships with others.	5.90	33.8	32.4	27.9	2.82 ± 0.91	
	4	Spend time with close friends.	0.00	23.5	30.9	45.6	3.22 ± 0.80	
	5	Find it easy to show concern, love, and warmth to others.	7.40	25.0	51.5	16.2	2.76 ± 0.81	
	6	Touch and am touched by people I care about.	0.00	27.9	38.2	33.8	3.22 ± 0.80	
	7	Find ways to meet my needs for intimacy.	2.90	39.7	48.5	8.80	2.63 ± 0.68	
	8	Settle conflicts with others through discussion and compromise.	5.90	36.8	47.1	10.3	2.62 ± 0.75	
	9	Seek guidance or counselling when necessary	4.40	52.9	25.0	17.6	2.56 ± 0.83	
nutrition (nine items)	1	Read labels to identify nutrient, fat, and sodium content in packaged food.	17.6	33.8	23.5	25.0	2.56 ± 1.056	2.53 ± 1.19 (Moderate)
	2	Choose a diet low in fat, saturated fat, and cholesterol.	11.8	57.4	16.2	14.7	2.34 ± 0.87	
	3	Limited use of sugars and food containing sugar (sweets).	11.8	48.5	25.0	14.7	2.43 ± 0.88	
	4	Eat 6–11 servings of bread, cereal, rice and pasta daily.	19.1	44.1	29.4	7.40	2.25 ± 0.85	
	5	Eat 2–4 servings of fruit each day.	5.90	44.1	38.2	11.8	2.56 ± 0.78	
	6	Eat 3–5 servings of vegetables each day.	5.90	5.00	30.9	13.2	2.51 ± 0.80	
	7	Eat 2–3 servings of milk, yogurt or cheese each day.	7.40	44.1	32.4	15.2	2.99 ± 3.66	
	8	Eat only 2–3 servings from the meat, poultry, fish, dried beans, eggs, and nuts group each day.	7.40	45.6	33.8	13.2	2.53 ± 0.82	
	9	Eat breakfast.	13.2	30.9	32.4	23.5	2.66 ± 0.98	
physical activity (eight items)	1	Follow a planned exercise program.	10.3	44.1	29.4	16.2	2.51 ± 0.88	2.52 ± 0.97 (Moderate)
	2	Exercise vigorously for 20 or more minutes at least three times a week (such as brisk walking, bicycling, aerobic dancing, using a stair climber).	8.80	42.6	32.4	16.2	2.56 ± 0.87	
	3	Take some time for relaxation each day.	7.40	26.5	36.8	29.4	2.88 ± 0.92	
	4	Take part in light to moderate physical activity (such as sustained walking 30–40 min 5 or more times a week).	8.80	44.1	27.9	19.1	2.57 ± 0.90	
	5	Take part in leisure-time (recreational) physical activities (such as swimming, dancing, bicycling).	7.40	51.5	23.5	17.6	2.51 ± 0.87	
	6	Do stretching exercises at least 3 times per week.	17.6	51.5	20.6	10.3	2.24 ± 0.86	
	7	Get exercise during usual daily activities (such as walking during lunch, using stairs instead of elevators, parking car away from the destination and walking).	8.80	42.6	26.5	22.1	2.62 ± 0.93	
	8	Balance time between work and play.	5.90	41.2	36.8	16.2	2.63 ± 0.82	
spiritual growth (nine items)	1	Feel I am growing and changing in positive ways.	2.90	32.4	42.6	22.1	2.86 ± 0.80	2.74 ± 0.82 (High)
	2	Believe that my life has a purpose.	1.50	29.4	35.3	33.9	3.01 ± 0.83	
	3	Look forward to the future.	0.00	19.1	38.2	42.6	3.24 ± 0.75	
	4	Feel content and at peace with myself.	2.90	29.4	48.5	19.1	2.84 ± 0.76	
	5	Work toward long-term goals in my life.	4.40	36.8	27.9	30.9	2.85 ± 0.91	
	6	Practice relaxation or meditation for 15–20 min daily.	19.1	52.9	19.1	8.8	2.18 ± 0.84	
	7	Am aware of what is important to me in life.	5.90	35.3	32.4	26.5	2.79 ± 0.90	
	8	Pace myself to prevent tiredness.	10.3	48.5	39.7	1.50	2.32 ± 0.67	
	9	Feel connected with some force greater than myself.	8.80	35.3	36.8	19.1	2.66 ± 0.89	
stress management (eight items)	1	Get enough sleep.	5.90	52.9	27.9	13.2	2.49 ± 0.80	2.51 ± 0.83 (Moderate)
	2	Accept those things in my life which I cannot change.	5.90	32.4	45.6	16.2	2.72 ± 0.80	
	3	Concentrate on pleasant thoughts at bedtime.	4.40	35.3	47.1	13.2	2.69 ± 0.75	
	4	Use specific methods to control my stress.	14.7	36.8	35.3	13.2	2.47 ± 0.90	
	5	Find each day interesting and challenging.	0.00	52.9	35.3	11.8	2.59 ± 0.69	
	6	Check my pulse rate when exercising.	25.0	47.1	13.2	14.7	2.18 ± 0.97	
	7	Reach my target heart rate when exercising.	26.5	39.7	26.5	7.4	2.15 ± 0.90	
	8	Expose myself to new experiences and challenges	2.90	35.3	38.2	23.5	2.82 ± 0.82	
Score low (1.60–2.25), moderate (2.26–2.71), and high (2.72–3.27)								

**Table 3 medicina-59-00106-t003:** Recorded data analysis of the questionnaire General Self-Efficacy Scale (GSE).

Item		Not at All True %	Hardly True %	Moderately True %	Exactly True %	Minimum	Maximum	Mean	Total Value ± SD
**1.**	I can always manage to solve difficult problems if I try hard enough	1.5	23.5	38.2	36.8	1	4	3.10	0.813
**2.**	If someone opposes me, I can find the means and ways to get what I want.	7.4	42.6	38.2	11.8	1	4	2.54	0.800
**3.**	It is easy for me to stick to my aims and accomplish my goals.	4.4	32.4	41.2	22.1	1	4	2.81	0.833
**4.**	I am confident that I could deal efficiently with unexpected events.	7.4	27.9	39.7	25.0	1	4	2.82	0.897
**5.**	Thanks to my resourcefulness, I know how to handle unforeseen situations	2.9	32.4	48.5	16.2	1	4	2.78	0.750
**6.**	I can solve most problems if I invest the necessary effort.	4.4	23.5	36.8	35.3	1	4	3.03	0.880
**7.**	I can remain calm when facing difficulties because I can rely on my coping abilities.	11.8	39.7	27.9	20.6	1	4	2.57	0.951
**8.**	When I am confronted with a problem, I can usually find several solutions.	2.9	27.9	45.6	23.5	1	4	2.90	0.794
**9.**	If I am in trouble, I can usually think of a solution	0.0	23.5	44.1	32.4	2	4	3.09	0.748
**10.**	I can usually handle whatever comes my way.	1.5	33.8	39.7	25.0	1	4	2.88	0.802
**Total α coefficient =**								**28.52**	**0.826**

**Table 4 medicina-59-00106-t004:** Evaluation of dental plaque index Quigley Hein and API before and after the intervention.

	N	Minimum	Maximum	Mean	SD	*p*
Initial value of Quigley-Hein Index	68	0.00	2.16	1.089	0.546	0.000 *
The final value of the Quigley-Hein Index	68	0.00	2.16	0.741	0.474	
Initial value of API	68	0.00	100.00	60.178	27.666	0.000 *
The final value of API	68	10.00	100.00	37.161	16.207	

* ANOVA test, correlation is significant at the 0.05 level.

**Table 5 medicina-59-00106-t005:** Assessment of healthy attitudes and behaviors before and after the intervention.

	Variables	Initial Value %	Final Value %
Dental brushing frequency/day	1 tooth brushing/day	4.4	0.0
	2 tooth brushing/day	70.6	75.0
	3 tooth brushing/day	25.0	25.0
Type of toothbrushing	manual brushing	86.8	86.8
	electric brushing	13.2	13.2
Use of dental floss	no	47.1	26.5
	yes	52.9	73.5

**Table 6 medicina-59-00106-t006:** Multiple regression analysis of predictors for HPLP II subcategories and demographic characteristics.

		Unstandardized Coefficients		Standardized Coefficients		
	Predictor	B	SE	Beta	t Value	*p* Value
Health responsibility	Residence(urban)	0.026	0.018	0.128	2.256	0.005
	Female gender	0.042	0.012	0.092	1.723	0.032
Interpersonal relationships	Residence(urban)	0.020	0.010	0.126	2.198	0.019
	Female gender	−0.132	0.013	−0.112	2.926	0.023
Nutrition	Residence(urban)	0.044	0.026	0.135	1.103	0.037
	Female gender	0.052	0.018	0.182	1.824	0.075
Physical activity	Residence(urban)	0.010	0.059	0.105	2.277	0.036
	Female gender	−0.156	0.201	−0.188	−2.103	0.028
Spiritual growth	Residence(urban)	0.005	0.002	0.109	1.099	0.030
	Female gender	−0.152	0.055	−0.133	−1.942	0.026
Stress management	Residence(urban)	0.052	0.023	0.211	2.134	0.002
	Female gender	−0.196	0.066	−0.163	−3.023	0.001

**Table 7 medicina-59-00106-t007:** Correlations between GSES items and demographic characteristics.

	GSES Item	Residence	Gender
1.	I can always manage to solve difficult problems if I try hard enough.	−0.020	0.191
2.	If someone opposes me, I can find the means and ways to get what I want.	0.109	0.023 *
3.	It is easy for me to stick to my aims and accomplish my goals.	−0.146	0.039 *
4.	I am confident that I could deal efficiently with unexpected events.	−0.144	−0.011
5.	Thanks to my resourcefulness, I know how to handle unforeseen situations.	0.102	−0.016
6.	I can solve most problems if I invest the necessary effort.	0.024 *	0.014 *
7.	I can remain calm when facing difficulties because I can rely on my coping abilities.	0.114	−0.103
8.	When I am confronted with a problem, I can usually find several solutions.	0.020 *	0.087
9.	If I am in trouble, I can usually think of a solution.	−0.159	−0.122
10.	I can usually handle whatever comes my way.	0.075	−0.101

* Correlation is significant at the 0.05 level (2-tailed).

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
