# Peer review of "Predictors of Oral Health Behaviors among Dental Students"

_medicina, 2023, doi:10.3390/medicina59010106_

Round 1

Reviewer 1 Report

Thank you for allowing me to review this scientific article, the purpose of which was to determine the oral health behaviors of dental students.
The topic of the paper is interesting, but the implementation is poor. First of all, it is written in inadequate English, and although I am not a native speaker, I find it difficult to read.
Here are some of the corrections needed:
1. The abstract is muddled and inadequate; anyone reading it would not read the article.
2. How was the sample size calculated?
3. Numbers are rounded to the nearest decimal place.
4. Table 2: ani?? Age (years) mean (sd) (min, max).
5. What does it mean? Recommended toothbrushing technique - please explain.
6. Why is the result 2.53± 0.9 (SD) presented like this? You can then also write the mean value in front of the first number.
7. How was the normality distribution of the obtained results checked?
8. Why the GSES tool who consists of 10 items scored on a 4-point Likert scale is not shown same as results in Table 2.
9. The authors state in the results that multiple regression analysis of demographic variables (age, sex, residence, and oral hygiene habits) with a score of six subcategories was performed to determine the predictors (Table 6). - and the same is not shown in the table.
10. Why was no correlation or regression performed between the demographic data and the GSES.

The idea of the work is interesting, but the implementation is lacking.

Author Response

University of Medicine and Pharmacy Grigore T.Popa 

Faculty of Dentistry, Department of Surgery,

University Street no.16, Iasi, Romania

email: carina.balcos@umfiasi.ro

21.12.2022

             Dear Reviewer,

            I would like to thank you once again for the precisely done review.  It was a great experience to follow all the indications and comments that you gave us. We followed all your recommendations which increased the value of this manuscript and gave us additional, valuable knowledge.

Best regards,

Yours sincerely, 

on behalf of the authors 

BalcoÈ™ Carina

Response to Reviewer 1 Comments

  1. The abstract is muddled and inadequate; anyone reading it would not read the article.
  2. At your suggestion, I re-edited the abstract so that it is easy to read and attractive.

  1. How was the sample size calculated?

R: When the study was completed, we had 78 students enrolled in the V year, English section. Applying the calculation formula for a confidence level of p = 95%, z = 1.96, with a margin of error of 5%, a number of 65 students should participate in the study to be relevant from a statistical point of view. Since several students expressed their desire to participate in the study, we enrolled all those who wished.

  1. Numbers are rounded to the nearest decimal place.

R: I modified it according to your recommendations.

  1. Table 2: ani?? Age (years) mean (SD) (min, max).

R: I modified it according to your recommendations.

  1. What does it mean? Recommended toothbrushing technique - please explain.

R: The recommended brushing technique for the healthy adult population, is the BASS technique. I have included in the text information related to the brushing technique.

  1. Why is the result 2.53± 0.9 (SD) presented like this? You can then also write the mean value in front of the first number.

R: 2.53± 0.90 (SD) is the average value for the whole domain of interpersonal relationships, calculated to determine the degree of change that all selected individuals manifest in that domain.

  1. How was the normality distribution of the obtained results checked?

R: We use the Shapiro-Wilk test to determine normality distribution.

  1. Why the GSES tool who consists of 10 items scored on a 4-point Likert scale is not shown the same as the results in Table 2?

R: I modified the table so that the distribution on the Likert scale of the responses to the GSE questionnaire also appears

  1. The authors state in the results that multiple regression analysis of demographic variables (age, sex, residence, and oral hygiene habits) with a score of six subcategories was performed to determine the predictors (Table 6). - and the same is not shown in the table.

R: At your suggestion, I removed the information that did not correspond to the data in the table.

  1. Why was no correlation or regression performed between the demographic data and the GSES?

R: In table 7 we have shown the results of the correlations between the GSES items and the demographic factors.

Reviewer 2 Report

1. Abstract Section:Line 26 represent conclusions out of results, this should be avoided, in results section only findings need to be presented.

2. The conclusions drawn are not supportive to the objectives of the study.  Conclusion presented is very confusion to understand and interrogative.

3. Line 110-111 is not clear to understand what author exactly means to say

4. Line 123 to 125: Author mentioned about additional information about correct way of brushing technique was provided to students. In what ways this additional information was provided as the means of provision such as videos, lectures or demonstrations might influence the oral health behaviours among dental students.

5. Discussion is too vast and vague enough. Author has presented extensive results but the discussion section lacks discussing these results in relation to literature evidence.

6. Line 348 - 349, the author repeat's the statistical analysis interpretation again in the conclusion. This is not recommended as the conclusion should contain an inference out of the results. 

Author Response

University of Medicine and Pharmacy Grigore T.Popa 

Faculty of Dentistry, Department of Surgery,

University Street no.16, Iasi, Romania

email: carina.balcos@umfiasi.ro

21.12.2022

             Dear Reviewer,

            I would like to thank you once again for the precisely done review.  It was a great experience to follow all the indications and comments that you gave us. We followed all your recommendations which increased the value of this manuscript and gave us additional, valuable knowledge.

Best regards,

Yours sincerely, 

on behalf of the authors 

BalcoÈ™ Carina

Response to Reviewer 2 Comments

  1. Abstract Section: Line 26 represent conclusions out of results, this should be avoided, in results section only findings need to be presented.

R: At your suggestion, I removed from the abstract comment related to the desire for change.

  1. The conclusions drawn are not supportive to the objectives of the study.  Conclusion presented is very confusion to understand and interrogative.

R: At your suggestion, I modified the conclusions so that they correspond to the objectives of the study.

  1. Line 110-111 is not clear to understand what author exactly means to say.

R: At your suggestion, I have modified the text so that it is more clear: API and Hein Q data were collected following the clinical examination and used to clinically determine the change in healthy behavior.

  1. Line 123 to 125: Author mentioned about additional information about correct way of brushing technique was provided to students. In what ways this additional information was provided as the means of provision such as videos, lectures or demonstrations might influence the oral health behaviours among dental students.

R:  I modified it according to your recommendations.

  1. Discussion is too vast and vague enough. Author has presented extensive results but the discussion section lacks discussing these results in relation to literature evidence.

R: I modified it according to your recommendations.

  1. Line 348 - 349, the author repeat's the statistical analysis interpretation again in the conclusion. This is not recommended as the conclusion should contain an inference out of the results. R: I modified it according to your recommendations.

Round 2

Reviewer 1 Report

I please the authors to reduce the values of p, sd, mean to a maximum of 3 decimal places.

And I also ask that they discuss the obtained results in more detail. Also check english language once more.

Author Response

Dear reviewer,
We tried to make the changes recommended by you.

1. I please the authors to reduce the values of p, sd, mean to a maximum of 3 decimal places.
A: I have made the recommended corrections.

2. I also ask that they discuss the obtained results in more detail. Also, check the English language once more.
R: I have added more discussions related to the results obtained( table 4, lines 282-284, 303-304, 315-318, 326-328, 331-340, 

 The verification of the English translation was done by a native English speaker.

Sincerely,

Carina Balcos

Reviewer 2 Report

Congratulations and Good luck

Author Response

Dear Reviewer,

Thank you very much for your guidance. Happy holidays.

Sincerely,

Carina Balcos